# Study of Ce, Ca, Fe, and Mn-Doped LaCoO_3_ Perovskite Oxide for the Four-Way Purification of PM, NO_x_, CO, and HC from Diesel Engine Exhaust

**DOI:** 10.3390/ma15124149

**Published:** 2022-06-10

**Authors:** Yinghui Wang, Xiurong Guo, Danfeng Du, Shaochi Yang

**Affiliations:** 1Mechanical and Electrical Engineering Institute, Northeast Forestry University, Harbin 150040, China; sjidan@163.com (Y.W.); sc_yang_97@163.com (S.Y.); 2School of Electrical Engineering, Sui Hua University, Suihua 152061, China; 3Transportation College, Northeast Forestry University, Harbin 150040, China; dudfeng@nefu.edu.cn

**Keywords:** perovskite-type catalyst, diesel exhaust, sol-gel method, four-way purification, doping amount

## Abstract

Perovskite-type catalysts were widely used in the field of automobile exhaust purification due to their inherent physicochemical properties and excellent doping characteristics. A series of La_1−x_M_x_Co_1−y_N_y_O_3_ (M = Ce, Ca; N = Fe, Mn) perovskite-type catalyst samples were prepared by sol-gel method for the four-way purification of PM, NO_x_, CO, and HC emitted by diesel exhaust. The activity of catalyst samples was tested by simulation experiments and hydrogen temperature-programmed reduction (H_2_-TPR). Catalyst samples were characterized by means of XRD, FT-IR, SEM, BET, and XPS analysis. The results demonstrated that the perovskite-type catalyst samples with a particle pore size of 3–5 μm can be prepared by sol-gel method. When A-site of LaCoO_3_ perovskite-type oxide was doped by cerium ions, the catalyst samples produced small distortion. The doping of cerium ions to A-site was more conducive to the four-way purification of diesel exhaust than calcium ions. La_0.8_Ce_0.2_CoO_3_ perovskite-type samples showed the best purification efficiency, and the purification efficiencies of PM, NO_x_, CO, and HC were 90%, 85%, 94%, and 100%, respectively. When the B-site of La_0.8_Ce_0.2_CoO_3_ perovskite was doped with iron ions, the purification efficiency of catalyst samples for PM and NO_x_ can be further enhanced. When the B-site of La_0.8_Ce_0.2_CoO_3_ perovskite was doped with manganese ions, the purification efficiency of the catalyst samples for PM can be further enhanced. It can be seen from the simulation experiments that La_0.8_Ce_0.2_Co_0.7_Fe_0.3_O_3_ perovskite was the best doping amount, and the purification efficiencies of PM, NO_x_, CO, and HC were 95%, 92%, 94%, and 100%, respectively.

## 1. Introduction

The pollutants emitted by diesel exhaust mainly include particulate matter (PM), nitrogen oxides (NO_x_), carbon monoxide (CO), and hydrocarbons (HC) [1]. Among them, the micron PM emitted from diesel engine is an important factor causing haze weather, which can seriously damage the human respiratory system, and even lead to the increase of lung cancer incidence rate. NO_x_ emission is the main cause of acid rain. The combination of CO and human hemoglobin will cause headache, syncope, and even endanger human life. HC causes serious damage to the human respiratory, nervous, and hematopoietic systems, which mixed with NO_x_ under light conditions may form photochemical smog [2,3,4]. Since this century, the application research of perovskite-type catalysts in the field of engine aftertreatment technology has been rapidly developed, reasonable doping and modification of perovskite-type catalysts can effectively purify the pollutants emitted by engine [5,6]. Hong Wang et al. [7] prepared three kinds of different perovskite-type catalysts by combining sol-gel method and impregnation method. The XRD and SEM characterization results proved that the catalysts can be uniformly supported on the zeolite surface. The substitution of La^3+^ at the A site by Cs^+^ can enhance the catalytic activity of the catalysts. Xuelei Mei et al. [8] synthesized the double perovskite-type La_2_NiB′O_6_ catalyst material with a three-dimensional ordered macroporous structure successfully by crystal template method. The purification efficiency and catalytic activity of LaNiO_3_ on PM can be improved by doping the B site with manganese, iron, cobalt, and copper ions, respectively. When the molar ratio of cobalt ions to nickel ions is 1:1, the purification efficiency of La_2_NiCoO_6_ on PM can reach 98.2%. Shiva Abedi et al. [9] synthesized LaFeO_3_, LaFe_0.7_Mn_0.3_O_3_, and LaMn_0.7_Fe_0.3_O_3_ nano-scale perovskite-type catalysts with different doping amount by sol-gel method, the CO was used as a reducing agent to selectively reduce NO through simulation experiments, in addition to studying the function of synthesized catalysts and finding the optimal catalyst in the CO-SCR process by XRD, SEM characterization methods, and chemical reaction kinetic model. The results showed that the conversion efficiency of CO and NO increased with the synergistic effect of manganese ions and iron ions. When the stoichiometry of iron and manganese ions were 0.3 and 0.7, respectively, the catalyst activity can reach the highest, and the conversion efficiency of CO and NO improved with the increase of reaction temperature and decrease of space velocity. Fan Fang et al. [10] synthesized porous perovskite-type La_0.6_Sr_0.4_CoO_3-δ_ nanotubes by sol-gel method combined with electrospinning technique following the calcination, which were treated with nitric acid to obtain a catalyst with a larger specific surface area. The catalysts were tested by temperature-programmed experiments. The results showed that the catalytic activity of La_0.6_Sr_0.4_CoO_3-δ_ nanotubes for PM was positively correlated with its specific surface area. The research team also analyzed the specific purification mechanism. Jiang, Q et al. [11] doped the A site and B site of LaNiO_3_ perovskite with calcium and copper ions, respectively, and detailed the effect of different doping amount on the physicochemical structure of the catalyst materials, revealed that the doped perovskite-type catalysts can effectively promote O^2−^ transport during the redox reaction of HC, and can be used as an effective carrier for O_ads_ in the low-temperature oxidation reaction, while decreasing the activation temperature of the overall reaction. E. Magnone et al. [12] studied the effects of different synthesis methods of La_0.6_Sr_0.4_CoO_3-δ_ powder on the adsorption and desorption of oxygen, particle size, specific surface area of the catalytic materials, the results showed that different synthesis methods directly affect the particle size and specific surface area of the catalysts, and indirectly affect the adsorption and desorption capacity of the catalysts for oxygen. Long Tang et al. [13] prepared LaKCoO_3_/γ-Al_2_O_3_/cordierite catalysts by a two-step impregnation method. The porous structure of γ-Al_2_O_3_ can evenly distribute LaKCoO_3_ on the surface of cordierite, and obtain a high specific surface area, thermal stability, and catalytic activity. When the reaction temperature is 390 °C, the conversion rate of LaKCoO_3_/γ-Al_2_O_3_/cordierite catalyst for soot particles can reach 90%, and the CO_2_ selectivity is 99.8%. Siran Zhang et al. [14] proposed a method of constructing highly dispersed Pt and oxygen vacancies on the catalysts. The LaCo_x_Ni_0.87−x_Pt_0.13_O_3_ perovskite-type catalyst was supported on SiO_2_ by the citric acid method and the impregnation method. Under low-temperature conditions, the LaCo_x_Ni_1−x_O_3_ surface can form high-dispersion double active centers, namely Pt clusters and O vacancies, which showed significant activity for CO oxidation.

Although the diesel engine exhaust purification technology by perovskite-type catalysts has been developed greatly, most of them focused on the modification research of the A site or the B site. There are few studies about four-way purification of PM, NO_x_, CO, and HC emitted from diesel exhaust by modifying perovskite-type catalysts. A large number of studies about perovskite-type catalysts showed that the doping of the A site can help to increase the activity of the catalyst lattice oxygen and make the B site ions exist in multiple valence states, causing the increases of oxygen vacancy in the catalyst, which reduced the ignition temperature of the oxidation reaction. The A site of perovskite-type LaCoO_3_ is the ideal adsorption site for NO_x_ storage. which can be doped by alkali metals, alkaline earth metals or rare earth elements. while the doping of the B site directly affects the activity of the perovskite-type catalyst, which is the redox site of the purification process, it can be doped by transition metal elements [15,16]. The doped A or B site with other cations can rationally tune the physicochemical characteristics of perovskites, including redox performance, oxygen mobility, and ionic conductivity [17]. A large number of studies showed that the catalytic activity of perovskite was comparable to that of noble metal catalysts [18,19]. Therefore, new non-precious metal-doped perovskite-type catalysts can be developed.

In this paper, the A site of the perovskite-type LaCoO_3_ is doped with cerium and calcium ions, and the B site is doped with iron and manganese ions. The objective of this work was to evaluate the effect of A site (cerium and calcium ions) and B site (iron and manganese ions) substitution on the structural and purification properties of perovskite-type catalyst samples. The CO, HC, and PM emitted by diesel engines are utilized as NO_x_ reducing agents to achieve the purpose of purifying CO, HC, NO_x_, and PM simultaneously. Perovskite-type catalyst samples particles La_1−x_M_x_Co_1−y_N_y_O_3_ with different doping amount were prepared by the sol-gel method [20,21], the prepared catalyst samples were characterized by means of XRD, FT-IR, SEM, BET, and XPS analysis. Meanwhile, by related simulation experiments and H_2_ temperature-programmed reduction (H_2_-TPR) experiments, the influence factors of the four-way purification efficiency were analyzed from the perspective of purification mechanism.

## 2. Materials and Methods

### 2.1. Sample Preparation

Perovskite-type catalyst samples with formula of La_1−x_M_x_Co_1−y_N_y_O_3_ (M = Ce, Ca; N = Fe, Mn; x = 0, 0.1, 0.2, 0.3, 0.4, 0.5, 0.6; y = 0, 0.1, 0.2, 0.3, 0.4, 0.5, 0.6) were prepared by sol-gel method. La(NO_3_)_3_·6H_2_O, Co(NO_3_)_2_·6H_2_O, Ce(NO_3_)_3_·6H_2_O, Ca(NO_3_)_2_·6H_2_O, Mn(NO_3_)_2_·4H_2_O, Fe(NO_3_)_3_·9H_2_O, citrate acid, and deionized water were purchased from Shanghai Aladdin Biochemical Technology Co., Ltd., Shanghai, China. All the chemicals are analytical grade. Suitable amounts of chemicals based on stoichiometry were dissolved in deionized water to get a sol and the obtained solution was stirred vigorously using magnetic stirrer (JXX1-85-2, stirring speed 100–2000 r/min, Changzhou, China). The solution was heated on a water-bath pot (HH-1, control temperature 25–100 °C, accuracy ± 1.5 °C, Tsingtao, China); when the temperature of the solution was raised to 70 °C, a suitable amount of citric acid (the molar ratio of citric acid to metal ions is 1.2–1.5) was added and temperature was justified at 80 °C to evaporate to dryness with vigorous stirring. During the dehydration process, a poly condensation reaction carried out between nitrate ions and citric acid leading to formation of gel, the obtained wet gel was placed in a constant temperature drying box (WGL-125B, control temperature 25–300 °C, Tianjin, China) at 110 °C for 10 h to form a fluffy solid. The obtained bulk solid was placed in a Tube furnace (TL1200 Mini, Rated temperature 1100 °C, Guangzhou, China) calcined at 700 °C for 5 h and was cooled slowly to room temperature. Finally, the powders were finely ground to obtain the La_1−x_M_x_N_y_Co_1−y_O_3_ perovskite-type catalyst samples. The preparation process route of the catalyst samples is shown in Figure 1.

### 2.2. Characterization

The phase structure state, and ionic valence state of the catalyst crystal have a significant impact on the catalytic performance of the catalyst samples [22,23]. The phase composition of catalyst samples was characterized by an X-ray diffractometer (XRD; XRD-6100; Shimadzu, Japan) with monochromator in the continuous mode in the range 2θ = 20~80°, using Cu Kα radiation, with a scanning rate of 8°·min^−^^1^ and time per step of 0.5 s. The fourier transform infrared spectrometer (FTIR; Spectrum 400; PerkinElmer, Waltham, MA, USA) as utilized to characterize the chemical composition of catalyst samples, using diffuse reflection method to test the samples that were prepared by mixing crushed catalyst samples and KBr with a mass ratio about 1:100. The number of scans was 80 and the resolution was set to 4 cm^−1^. The Scanning Electron Microscope (SEM; Quanta 200; FEI, Valley City, ND, USA) utilized to characterize the microscopic morphology of catalyst samples. The samples were mounted on the aluminum holder with conductive adhesive and then sputtered with gold. Then the microscopic morphology of the samples was observed under the acceleration voltage of 13 kV. The N_2_ adsorption isotherms and pore size distribution (PSD) curves of perovskite-type catalyst samples were measured by Automatic Surface Area and Porosity Analyzer (ASAP 2020). XPS analysis was performed on an Amicus spectrometer equipped with Mg Kα X-ray radiation.

### 2.3. Experimental System and Methods

A simulation test bench for activity tests of catalyst samples is shown in Figure 2. The simulated exhaust gas can be provided by the gas cylinder group 2 (O_2_, N_2_, C_2_H_4_, CO, and NO) and mixed by the gas mixing chamber 14. The tube furnace 10 was used to control the temperature of the mixed gas and the water-bath pot 6 can maintain the exhaust temperature in the Quartz tube 7. According to reference [24], the exhaust gas temperature of diesel engine is allowed to vary from 550 °C to 650 °C. Therefore, the simulated exhaust gas temperature can be maintained at 600 °C by adjusting the tube furnace. The cordierite was used to support PM and catalyst samples, the mass of supported PM and catalyst samples was 18 mg, the mass ratio of PM to catalyst samples was 1:1, the supporting process of PM and catalyst samples is shown in Figure 3. The moderate PM and catalyst samples were set in the circulating blower, which was sealed before the supporting process. When the circulating blower started, the porous structure of cordierite can capture all of the powder PM and catalyst samples. The cordierite supported with PM and catalyst samples was placed inside the quartz tube 7, the length of quartz tube is 200 mm, which is twice that of cordierite, and their diameter is 80 mm, the gas analyzer (Horida Ltd., MEXA-4000FT, relative error 1.7%, Kyoto, Japan) was used to detect the concentration of gas generated after purification, the electronic balance (PWN224ZH, repeatability deviation 0.01 mg, Shanghai, China) was used to measure the mass of cordierite before and after purification.

All the experiments were carried out under the condition of room temperature and shading. The simulated gas was prepared by mixing O_2_, N_2_, C_2_H_4_, CO. and NO, and the gas ratio concentration was strictly controlled by the throttle valve and the flowmeter (DMF-1-S6, accuracy ± 1%), the gas flow rate of O_2_, N_2_, C_2_H_4_, CO, and NO can be controlled to 10 mL/min, 89.81 mL/min, 0.3 mL/min, 0.8 mL/min, and 0.8 mL/min separately. And the total flow rate was set to 10 mL/min, the concentration of simulated exhaust gas was shown in Table 1 [25,26]. The space velocity was $8.5\times 10^{4}\;h^{-1}$. When the experiment starts, the simulation test bench works stably for 10 min, according to the experimental requirements firstly. Then, experimental results were recorded and the test flow chart is shown in Figure 4. The conversion of CO, HC, NO_x_, and PM were calculated as follows:
(1)$$\eta_{CO}=\frac{{C^{in}}_{CO}-{C^{out}}_{CO}}{{C^{in}}_{CO}}$$
(2)$$\eta_{HC}=\frac{{C^{in}}_{HC}-{C^{out}}_{HC}}{{C^{in}}_{HC}}$$
(3)$$\eta_{NOx}=\frac{{C^{in}}_{NOx}-{C^{out}}_{NOx}}{{C^{in}}_{NOx}}$$
(4)$$\eta_{PM}=\frac{{M^{B}}_{PM}-{M^{A}}_{PM}}{{M^{B}}_{PM}}$$
where $\eta_{CO}$ is the purification efficiency of *CO*; ${C^{in}}_{CO}$ and ${C^{out}}_{CO}$ are concentration of before and after the experiment, ppm; $\eta_{HC}$ is the purification efficiency of *HC*; ${C^{in}}_{HC}$ and ${C^{out}}_{HC}$ are concentration of before and after the experiment, ppm; $\eta_{NOx}$ is the purification efficiency of *NO_x_*; ${C^{in}}_{NOx}$ and ${C^{out}}_{NOx}$ are concentration of before and after the experiment, ppm; $\eta_{PM}$ is the purification efficiency of *PM*; ${M^{B}}_{PM}$ and ${M^{A}}_{PM}$ are the mass of the filter before and after the experiment, mg.

## 3. Results and Discussion

### 3.1. Results of Characterization on Catalyst Samples Doping at A Site

X-ray diffraction patterns of catalyst samples La_1−x_M_x_CoO_3_ (M = Ce, Ca; x = 0, 0.1, 0.2, 0.3, 0.4, 0.5, 0.6) are shown in Figure 5. As can be observed from Figure 5a, perovskite diffraction patterns were observed for LaCoO_3_, La_0.9_Ce_0.1_CoO_3_, La_0.8_Ce_0.2_CoO_3_, and La_0.7_Ce_0.3_CoO_3_. These samples have characteristic diffraction peaks of LaCoO_3_ near 2θ = 23.3°, 33.0°, 42.4°, and 47.0°, and they all correspond to the perovskite-type cubic crystallized (PDF# 97-002-8921). There are CeO_2_ (PDF# 97-002-8709), Co_3_O_4_ (PDF# 00-009-0418) impurity phases in the samples. Therefore, when Ce-content > 0.3, the XRD characteristic diffraction peaks of LaCoO_3_ near 2θ = 33.0° was extraordinarily inconspicuous. According to reference [27], the effective ionic radius of lanthanum ions and cerium ions are 1.10 Å and 0.87 Å, respectively, therefore, excessive cerium ions and cobalt ions cannot be completely dissolved in the LaCoO_3_ lattice. As can be observed from Figure 5b, perovskite diffraction patterns were observed for La_0.9_Ca_0.1_CoO_3_, La_0.8_Ca_0.2_CoO_3_, La_0.7_Ca_0.3_CoO_3_, La_0.6_Ca_0.4_CoO_3_, La_0.5_Ca_0.5_CoO_3_, and La_0.4_Ca_0.6_CoO_3_. All samples have characteristic diffraction peaks of LaCoO_3_, and they all correspond to the perovskite-type cubic crystallized (PDF# 97-002-8921). The effective ionic radius of calcium ions is 1.06 Å, which is close to the ion radius of lanthanum ions, therefore, there are only small amounts of CaCO_3_ (PDF# 97-002-8827) and Co_3_O_4_ (PDF# 00-009-0418) in the impurity phase in the crystal samples. Through the above analysis, it can be concluded that the structure of La_0.8_Ce_0.2_CoO_3_ and La_0.8_Ca_0.2_CoO_3_ perovskite-type catalyst samples reaches the best catalyst conditions. As can be observed from Figure 5a,b, the characteristic diffraction peaks of LaCoO_3_ doping with cerium ions and calcium ions near 2θ = 23.3° moved left slightly. This information would confirm the accommodation of these compounds within the perovskite lattice.

Figure 6 shows the FT-IR spectra of La_0.8_Ce_0.2_CoO_3_ and La_0.8_Ca_0.2_CoO_3_ catalyst samples. As can be observed from Figure 6, there are several vibration bands at 424, 500, 596, 668, 712, 875, 1415, 1460, and 1628 cm^−1^. FT-IR peaks located below 1000 cm^−1^ has been reported to represent metal oxides [28]. The vibration band at 424 cm^−1^ and 596 cm^−1^ belongs to the bending vibration of Co–O bonding in the BO6 octahedron, and the vibration band at 500 and 669 cm^−1^ belongs to the bending vibration of Co–O and Ce–O bonding, which are attributed to Co_3_O_4_ and CeO_2_ [29,30]. The absorption peaks of 875 and 712 cm^−1^ belong to the calcite crystals, which are related to the bending vibration of C–O bond. The absorption peaks appeared at 1415, 1460, and 1628 cm^−1^ of catalyst samples, representing the bending mode of C–H. Compared to FTIR spectra of La_0.8_Ca_0.2_CoO_3_ and La_0.8_Ce_0.2_CoO_3_ catalyst samples, in the FTIR spectra of La_0.8_Ce_0.2_CoO_3_, the band at 596 cm^−1^ becomes broad and up-shifting, suggesting that some amounts of Co^3+^ changed to Co^2+^ when some La^3+^ was replaced by Ce^3+^ and Ce^4+^. However, there is no change in the valence state of Ca^2+^, so there is no other absorption spectrum in the FTIR spectra of La_0.8_Ca_0.2_CoO_3_. After the FTIR analysis, it can be confirmed that some impurities like carbonate group and hydroxyl group presented in samples.

### 3.2. Results of Simulation Experiments on Catalyst Samples Doping at A Site

MATLAB software was used to fit the experimental data. Figure 7a–d show the time curve of the purification efficiency of NO_x_, PM, HC, and CO when the A site was doped with different concentrations of cerium ions. As can be observed from Figure 7, when the purification time does not exceed 90 s, the purification efficiency increases with time. At this time, the catalyst samples are in the heating stage and do not exert the best catalytic performance. When the time exceeds 90 s, the purification efficiency reaches the highest. When the purification time exceeds 120 s, the purification efficiencies of NO_x_ and PM show an obvious downward trend, which can be attributed to the decrease of PM load with purification time increases, decreasing the NO_x_ and PM redox reaction rate. When the doping amount does not exceed 0.2, the purification efficiencies of NO_x_, PM, HC, and CO improve with the increase of the doping amount. When x = 0.2, the purification efficiency can reach the highest, which can reach 85%, 90%, 94%, and 100%, respectively. When x > 0.3, excessive cerium ions were not doped in perovskite-type oxide, too much impurity phase was introduced into the catalyst samples. These impurities have poor purification efficiency on NO_x_, PM, HC, and CO, which are consistent with the results of XRD.

Figure 8a–d show the time curve of the purification efficiency of NO_x_, PM, HC, and CO when the A site was doped with different concentrations of calcium ions. Upon comparison of perovskite-type catalyst samples doped with calcium ions and cerium ions, the purification trend and efficiency of PM, HC, and CO are basically the same. However, the doping of calcium ions shows a lower efficiency in purifying NO_x_, and the highest purification efficiency is 72%, which can be attributed to the factor that the valence state of the doped calcium ions has not changed, and the Co^3+^ in the catalyst samples change to Co^4+^, but not to Co^2+^. The existence of Co^2+^ is more conducive to the formation of O vacancies and adsorbed NO, which is beneficial to the reduction of NO, and the change of ions valence state promotes the adsorption and desorption of O. In addition, When A-site of LaCoO_3_ perovskite was doped with cerium ions, the catalyst crystal produces small distortion and enhances the purification efficiency of PM, NO_x_, CO, and HC, which is consistent with the research content reported in the literature [3,4,5,6,7,8,9,10,11,12,13,14,15,16,17,18,19,20,21,22,23,24,25,26,27,28,29,30,31]. According to the literature [32,33,34,35,36] on the mechanism analysis of purifying the exhaust pollutants by perovskite-type catalysts, combined with the experiment results of this study, the overall purification and purification process is inferred as Formulas (5)–(19).
(5)$$\mathrm{NO}↔{\mathrm{NO}}_{ads}$$
(6)$$\mathrm{CO}↔{\mathrm{CO}}_{\begin{array}{l} ads \end{array}}$$
(7)$${2N}_{ads}↔N_{2}$$
(8)$$O_{\begin{array}{l} 2 \end{array}}↔2O_{ads}$$
(9)$$C_{2}H_{4}↔C_{2}H_{3ads}+H_{ads}$$
(10)$${\mathrm{NO}}_{ads}\to N_{ads}+O_{ads}$$
(11)$${\mathrm{CO}}_{\begin{array}{l} ads \end{array}}+O_{ads}\to {\mathrm{CO}}_{\begin{array}{l} 2 \end{array}}$$
(12)$${\mathrm{PM}+O}_{ads}\to {\mathrm{CO}}_{\begin{array}{l} 2 \end{array}}+{\mathrm{CO}}_{}$$
(13)$${\mathrm{NO}}_{ads}+O_{ads}\to {\mathrm{NO}}_{2}$$
(14)$$\mathrm{PM}+{\mathrm{NO}}_{2ads}\to \mathrm{CO}+{\mathrm{CO}}_{\begin{array}{l} 2 \end{array}}+N_{2}$$
(15)$${\mathrm{CO}}_{\begin{array}{l} ads \end{array}}{+\mathrm{NO}}_{2ads}\to {\mathrm{CO}}_{\begin{array}{l} 2 \end{array}}+N_{2}$$
(16)$$C_{2}H_{3ads}{+\mathrm{NO}}_{2ads}\to {\mathrm{CO}}_{\begin{array}{l} 2 \end{array}}+N_{2}+H_{2}O$$
(17)$$H_{ads}{+\mathrm{NO}}_{2ads}\to N_{2}+H_{2}O$$
(18)$$H_{ads}{+\mathrm{NO}}_{ads}\to N_{2}+H_{2}O$$
(19)$$C_{2}H_{3ads}{+\mathrm{NO}}_{ads}\to {\mathrm{CO}}_{\begin{array}{l} 2 \end{array}}+N_{2}+H_{2}O$$

The deconvolution of O 1 s signals was conducted in order to estimate the species in detail. As shown in Figure 9, The O 1 s spectrum for La_0.8_Ca_0.2_CoO_3_ and La_0.8_Ce_0.2_CoO_3_ could be decomposed into two components. The peak at about 529.1 eV was ascribed to the lattice oxygen species (O*_latt_*). Another component at 531.2 eV was assigned to adsorbed oxygen species (O*_ads_*) [37], which has been reported as active species in oxidation reactions due to the electrophilic nature [38]. Since the surface O*_ads_*/O*_latt_* molar ratio of the La_0.8_Ce_0.2_CoO_3_ sample (0.78) was higher than that of the La_0.8_Ca_0.2_CoO_3_ sample (0.62). Therefore, the La_0.8_Ce_0.2_CoO_3_ sample showed the better purification efficiency in NO, which are consistent with the reaction pathway reported in Equations (5)–(19).

The SEM micrographs clearly show large differences in the micro-structure and morphology of La_0.8_Ce_0.2_CoO_3_ and La_0.8_Ca_0.2_CoO_3_ perovskite-type catalyst samples from Figure 10a,b, La_0.8_Ce_0.2_CoO_3_ perovskite-type catalyst samples represent porous morphologies, La_0.8_Ca_0.2_CoO_3_ perovskite-type catalyst samples represent spongy morphologies. Both samples are uniformly dispersed without sintering. The porous structure of La_0.8_Ce_0.2_CoO_3_ perovskite-type catalyst samples are clearer than La_0.8_Ca_0.2_CoO_3_, the pore size of which ranging from 3 µm to 5 µm.

### 3.3. Results of Characterization on Catalyst Samples Doping at B Site

On the basis of the above research results, the A-site doping amount of cerium ions was selected, and the doping amount was 0.2. The B-site doping of manganese and iron ions on the catalytic performance of the La-based perovskite-type catalyst should be further studied. According to reference [27], the effective ionic radius of iron ions, cobalt ions and manganese ions are 0.645 Å, 0.61 Å, and 0.645 Å, respectively, which meet the 15% rule. Therefore, La_0.8_Ce_0.2_Co_1−y_N_y_O_3_ (N = Fe, Mn; y = 0, 0.1, 0.2, 0.3, 0.4, 0.5, 0.6) can form continuous series of solid solutions.

X-ray diffraction patterns of catalyst samples La_0.8_Ce_0.2_Co_1−y_N_y_O_3_ (N = Fe, Mn; y = 0, 0.1, 0.2, 0.3, 0.4, 0.5, 0.6) are shown in Figure 11. As can be observed from Figure 11a, perovskite diffraction patterns can be observed for La_0.8_Ce_0.2_Co_0.9_Fe_0.1_O_3_, La_0.8_Ce_0.2_Co_0.8_Fe_0.2_O_3_, La_0.8_Ce_0.2_Co_0.7_Fe_0.3_O_3_, La_0.8_Ce_0.2_Co_0.6_Fe_0.4_O_3_, La_0.8_Ce_0.2_Co_0.5_Fe_0.5_O_3_, and La_0.8_Ce_0.2_Co_0.4_Fe_0.6_O_3_. All samples have characteristic diffraction peaks of LaCoO_3_ near 2θ = 23.3°, 33.0°, 42.4°, and 47.0°, they all correspond to the perovskite-type cubic crystallized (PDF# 97-002-8921). There are CeO_2_ (PDF# 97-002-8709), FeO (PDF# 97-008-2236), and Fe_3_O_4_ (PDF# 00-026-1136), as well as a small amount of Co_3_O_4_ (PDF# 00-009-0418) impurity phases in the crystal samples. However, the doping of iron ions enhances the solubility of cerium ions in the crystal lattice. As can be observed from Figure 11b, perovskite diffraction patterns can be observed for La_0.8_Ce_0.2_Co_0.9_Mn_0.1_O_3_, La_0.8_Ce_0.2_Co_0.8_Mn_0.2_O_3_, La_0.8_Ce_0.2_Co_0.7_Mn_0.3_O_3_, and La_0.8_Ce_0.2_Co_0.6_Mn_0.4_O_3_, they all correspond to the perovskite-type cubic crystallized. There are amount of MnO_2_ (PDF# 00-053-0633), CeO_2_ (PDF# 97-002-8709), and Co_3_O_4_ (PDF# 00-009-0418) impurity phases in the crystal samples. LaCoO_3_ perovskite seems to accept a higher cobalt ions substitution degree by iron ions than manganese ions, without destabilizing the perovskite structure. In fact, the continuous series of solid solutions of atoms in perovskite crystals has not been realized. This may be due to the fact that cerium atoms cannot achieve continuous series of solid solutions in perovskite crystals. Through the above analysis, it can be concluded that the structure of La_0.8_Ce_0.2_Co_0.7_Fe_0.3_O_3_ and La_0.8_Ce_0.2_Co_0.7_Mn_0.3_O_3_ perovskite-type catalyst samples reaches the best catalyst conditions. As can be observed from Figure 11a,b, the characteristic diffraction peaks of La_0.8_Ce_0.2_CoO_3_ doping with cerium ions and calcium ions near 2θ = 33° moved right slightly. This information would confirm the accommodation of these compounds within the perovskite lattice.

Figure 12 shows the FT-IR spectra of La_0.8_Ce_0.2_Co_0.7_Fe_0.3_O_3_ and La_0.8_Ce_0.2_Co_0.7_Mn_0.3_O_3_ catalyst samples. As can be observed from Figure 12, there are several vibration bands at 424, 470, 500, 530, 543, 550, 561, 562, 596, 668, 1460, and 1628 cm^−1^ for La_0.8_Ce_0.2_Co_0.7_Fe_0.3_O_3_ and La_0.8_Ce_0.2_Co_0.7_Mn_0.3_O_3_ catalyst samples. The vibration band at 470 and 550 cm^−1^ belongs to the bending vibration of Fe–O bonding in the BO_6_ octahedron, and the peak at 561 cm^−1^ is attributed to asymmetrical modes of the Mn–O bond of MnO_6_ octahedrons groups of LaMnO_3_ perovskite. The vibration band at 543 and 560 cm^–1^ belongs to the bending vibration of Fe–O bonding in the FeO and Fe_3_O_4_ crystals, respectively. The vibration band at 530 cm^–1^ belongs to the bending vibration of Mn–O bonding in the MnO_2_ crystals [39,40]. Several vibration band of some impurities like carbonate group and hydroxyl group also can be observed from Figure 12, which would not be analysed in this paper.

### 3.4. Results of Simulation Experiments on Catalyst Samples Doping at B Site

Figure 13a–d shows the fitting curves of the purification efficiencies of NO_x_, PM, HC, and CO over time when the B site was doped with different concentrations of iron ions. As can be observed from Figure 13, when the purification time does not exceed 90 s, the purification efficiency increases with time. When the time exceeds 90 s, the purification efficiency reaches the highest. When the purification time exceeds 180 s, the purification efficiency of NO_x_ and PM shows an obvious downward trend. When y = 0.3, the purification efficiency can reach the highest. Compared with La_0.8_Ce_0.2_CoO_3_, the highest purification efficiency for PM and NO_x_ is increased by 5% and 7%, respectively, while the purification efficiency for HC and CO is hardly improved.

Figure 14a–d shows the time curve of the purification efficiency of NO_x_, PM, HC, and CO when the B site is doped with different concentrations of manganese ions. Comparison of perovskite-type catalysts doped with iron ions and manganese ions, the purification trend of NO_x_, PM, HC, and CO are basically the same. Compared with La_0.8_Ce_0.2_CoO_3_, when y = 0.3, the highest purification efficiency for PM and NO_x_ is increased by 6% and 2%, which can be speculated that the doping of manganese ions at the B site is more conducive to the oxidation reaction, and the doping of iron ions at the B site is more conducive to the reduction reaction.

The H_2_-TPR experiments were conducted on a TPDRO instrument (TP-5080, Xianquan, Tianjin, China) using a thermal conductivity detector (TCD). H_2_-TPR techniques were used to measure the reducibility of La_0.8_Ce_0.2_Co_0.7_Fe_0.3_O_3_ and La_0.8_Ce_0.2_Co_0.7_Mn_0.3_O_3_ catalyst samples. The corresponding H_2_-TPR profiles are presented in Figure 15. Both of the samples exhibit two major reduction peaks, the first reduction peak of La_0.8_Ce_0.2_Co_0.7_Fe_0.3_O_3_ at 340 °C can be attributed to the reduction of Fe^4+^ to Fe^3+^ and consumption of adsorbed oxygen [41]. The first reduction peak of La_0.8_Ce_0.2_Co_0.7_Mn_0.3_O_3_ at 339 °C can be attributed to the reduction of Mn^4+^ to Mn^3+^ consumption of adsorbed oxygen [42]. The second reduction peak of La_0.8_Ce_0.2_Co_0.7_Fe_0.3_O_3_ appears at 551 °C, which can be ascribed to the reduction of Fe^3+^ to Fe^2+^ and consumption of lattice oxygen [43]. The second reduction peak of La_0.8_Ce_0.2_Co_0.7_Mn_0.3_O_3_ appears at 762 °C, which can be ascribed to the reduction of Mn^3+^ to Mn^2+^ and consumption of lattice oxygen [44]. Through the analysis above, it can be concluded that the redox properties and the oxygen mobility of the catalysts are important influencing factors in purification efficiency of NO_x_, PM, HC, and CO.

The SEM micrographs of La_0.8_Ce_0.2_Co_0.7_Fe_0.3_O_3_ and La_0.8_Ce_0.2_Co_0.7_Mn_0.3_O_3_ perovskite-type catalyst crystal samples are shown in Figure 16a,b, La_0.8_Ce_0.2_Co_0.7_Fe_0.3_O_3_ perovskite-type catalyst samples are more uniformly dispersed, and La_0.8_Ce_0.2_Co_0.7_Mn_0.3_O_3_ perovskite-type catalyst samples are partially reunited. Both of the perovskite-type catalyst samples exist porous structure. The macropore with diameter ranging from 3 µm to 5 µm in perovskite-type catalyst samples. Perovskite-type catalyst crystal samples can contact the PM and hazardous gas sufficiently emitted from engine owing to their high specific surface area and porosity [45]. La_0.8_Ce_0.2_Co_0.7_Fe_0.3_O_3_ perovskite-type catalyst samples were more conducive to the application in the field of automobile exhaust purification.

Figure 17a,b show N_2_ adsorption isotherms and pore size distribution (PSD) curves of LaCoO_3_, La_0.8_Ce_0.2_Fe_0.3_Co_0.7_O_3_, and La_0.8_Ce_0.2_Co_0.7_Mn_0.3_O_3_ perovskite-type catalyst samples separately. Based on the International Union of Pure and Applied Chemistry classification (IUPAC), the isotherms curves in Figure 17a reveals an H3 hysteresis loop related to mesoporous or macroporous materials. It can be seen that the order of specific surface area is LaCoO_3_, La_0.8_Ce_0.2_Fe_0.3_Co_0.7_O_3_, and La_0.8_Ce_0.2_Co_0.7_Mn_0.3_O_3_. Significant increase in nitrogen adsorption at high relative pressure (0.9 < P/P_0_ < 1) and the PSD curves illustrate the extensive existence of the mesopores and macropore, which are well consistent with the analysis of SEM. Figure 17b reveals that La_0.8_Ce_0.2_Fe_0.3_Co_0.7_O_3_ perovskite -type catalyst sample has the largest pore volume of mesopores (mainly concentrated in 50 nm). The detailed textural parameters of catalyst samples are listed in Table 2. It is widely accepted that the impurities formation as a consequence of the limited accommodation of A and B cations leads to a progressive decrease of specific surface area [46]. Based on the above analysis, it can be further inferred that the impurity content of La_0.8_Ce_0.2_Co_0.7_Fe_0.3_O_3_ perovskite-type catalyst crystal samples is less than that of La_0.8_Ce_0.2_Co_0.7_Mn_0.3_O_3_ perovskite-type catalyst crystal samples, which are consistent with the results of XRD and FT-IR.

Based on the above analysis, the purification data for the best samples (x = 0.2; y = 0; 0.3) are shown in Table 3. It can be seen that appropriate doping of A and B sites is effective for the four-way purification efficiency. Combined with the analysis results of t SEM and BET, La_0.8_Ce_0.2_Co_0.7_Fe_0.3_O_3_ perovskite-type catalyst samples can be further proved to be the optimal doping concentration.

## 4. Conclusions

Several perovskite-type catalyst samples were developed for the four-way purification of diesel engine exhaust by doping A and B Sites in perovskite LaCoO_3_. The prepared perovskite-type catalyst samples were characterized by XRD, SEM, FT-IR, and BET analysis, and their activity were tested by simulation experiments. The following conclusions were reached:The porous perovskite structures can be observed from prepared perovskite-type catalyst samples. When the A-site and B-site doping amount of LaCoO_3_ exceeds a certain value, more impurity phases will be produced;Macropores of 3 µm to 5 µm are presented in La_0.8_Ce_0.2_Fe_0.3_Co_0.7_O_3_ and La_0.8_Ce_0.2_Co_0.7_Mn_0.3_O_3_ perovskite-type catalyst samples, and La_0.8_Ce_0.2_Fe_0.3_Co_0.7_O_3_ has the largest pore volume of mesopores (mainly concentrated in 50 nm);Doping the A site of LaCoO_3_ perovskite-type oxide can change the valence state of the B site ions to a certain extent, which is conducive to the occurrence of redox reactions;On the basis of cerium ions doping the A site of LaCoO_3_ perovskite-type oxide, the doping of manganese ions at the B site can further improve the purification ability on PM. However, the doping of iron ions at the B site can enhance the purification ability on PM and NO_x_; andLa_0.8_Ce_0.2_Co_0.7_Fe_0.3_O_3_ shows the best purification ability and the least impurity phase in the samples, the purification efficiency of PM, NO_x_, HC, and CO are 95%, 92%, 94%, and 100%, respectively.

## Figures and Tables

**Figure 1 materials-15-04149-f001:**
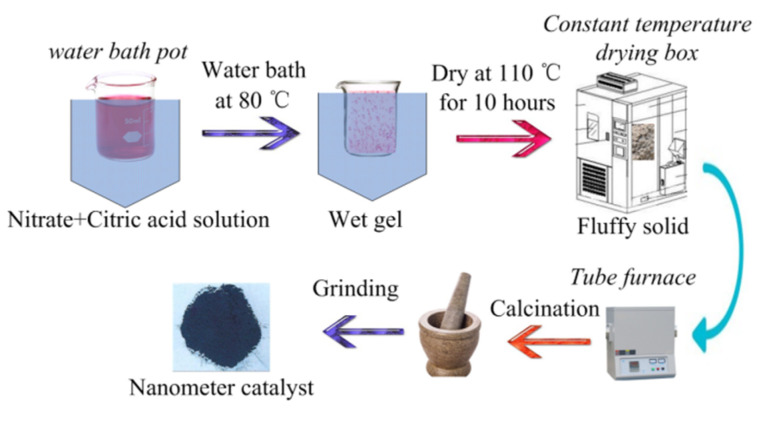
Preparation process route of the catalyst samples.

**Figure 2 materials-15-04149-f002:**
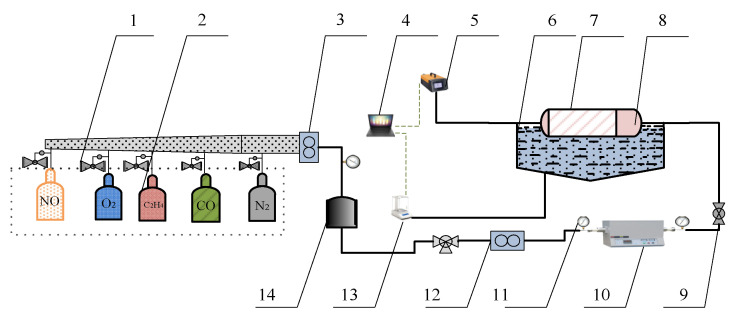
The overall structure of the simulation test bench. 1. Pressure-reducing valve. 2. Gas cylinder group. 3,12. Flowmeter. 4. Computer. 5. Gas analyzer. 6. Water bath pot. 7. Quartz tube. 8. Cordierite loaded with PM and catalytic agent. 9. Ball valve. 10. Tube furnace. 11. Pressure gauge. 13. Electronic balance. 14. Gas mixing chamber.

**Figure 3 materials-15-04149-f003:**
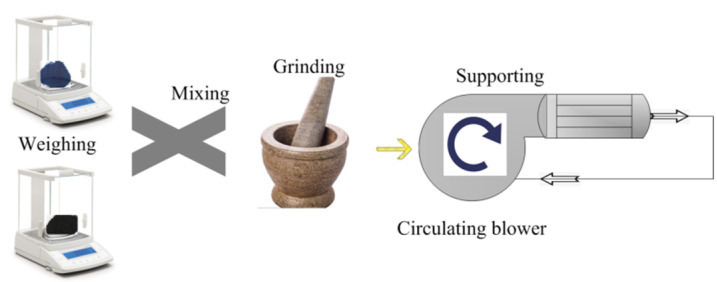
The supporting process of PM and catalyst samples.

**Figure 4 materials-15-04149-f004:**
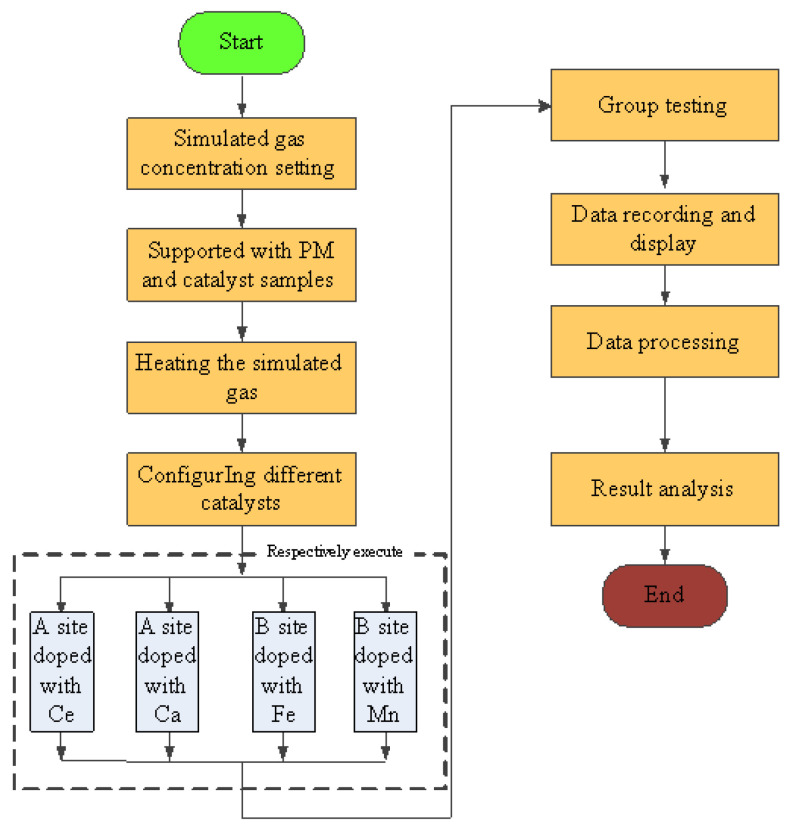
Simulation test flow chart.

**Figure 5 materials-15-04149-f005:**
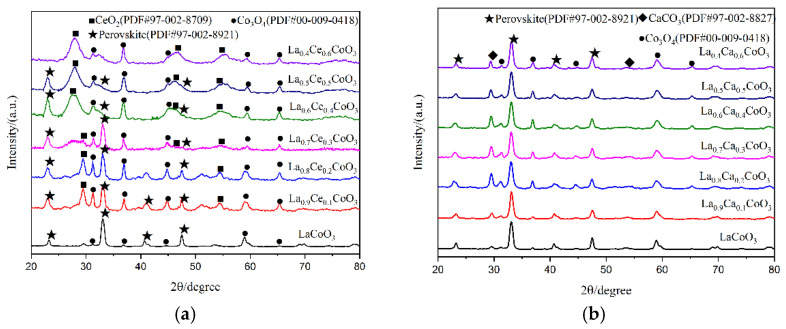
XRD patterns of catalyst samples: (**a**) La_1−x_Ce_x_CoO_3_ catalyst samples; and (**b**) La_1−x_Ca_x_CoO_3_ catalyst samples.

**Figure 6 materials-15-04149-f006:**
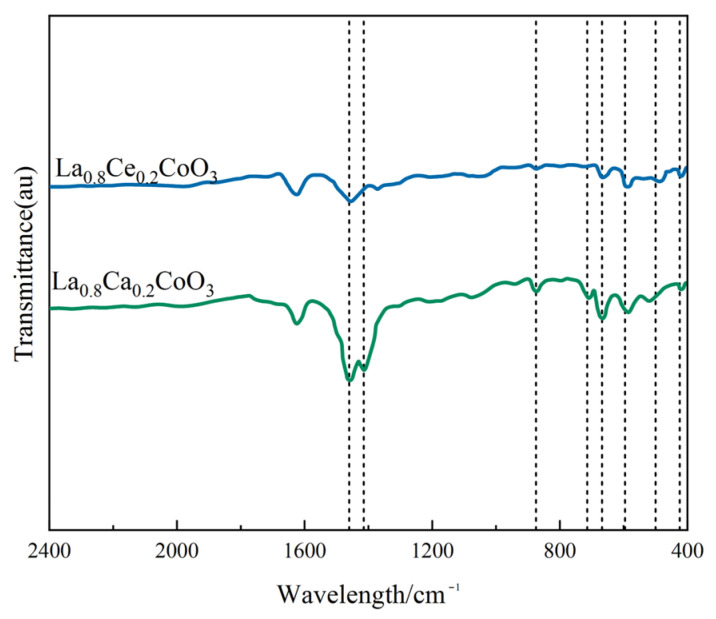
FTIR spectra of La_0.8_Ce_0.2_CoO_3_ and La_0.8_Ca_0.2_CoO_3_ catalyst samples.

**Figure 7 materials-15-04149-f007:**
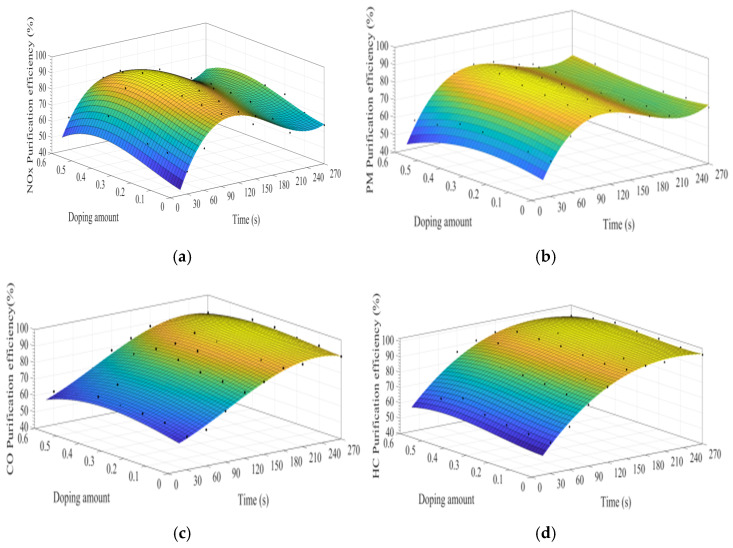
Purification efficiency curve of catalyst when A site was doped with ceriumions: (**a**) NO_x_; (**b**) PM; (**c**) CO; and (**d**) HC.

**Figure 8 materials-15-04149-f008:**
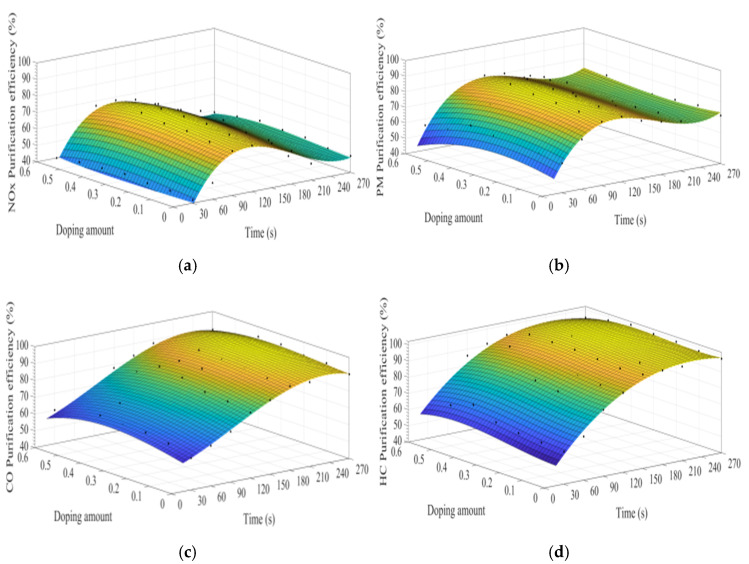
Purification efficiency curve of catalyst when A site was doped with calcium ions: (**a**) NO_x_; (**b**) PM; (**c**) CO; and (**d**) HC.

**Figure 9 materials-15-04149-f009:**
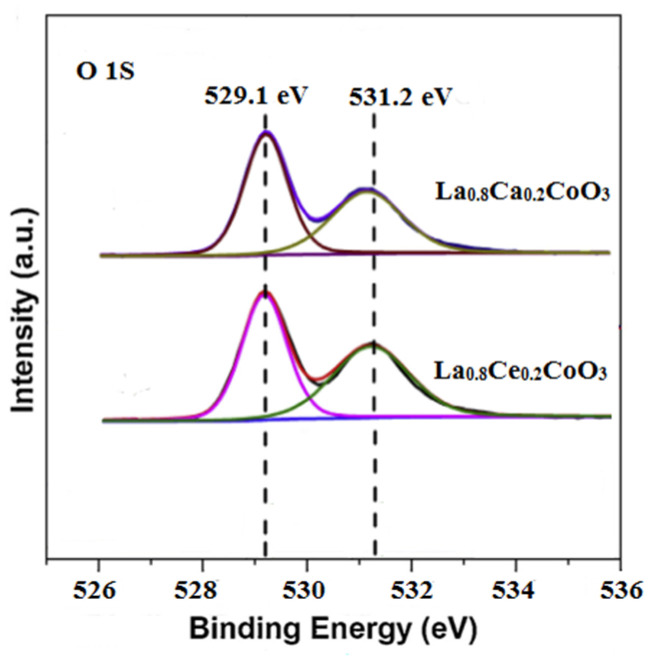
X-ray photoelectron of the O 1 s for La_0.8_Ca_0.2_CoO_3_ and La_0.8_Ce_0.2_CoO_3_.

**Figure 10 materials-15-04149-f010:**
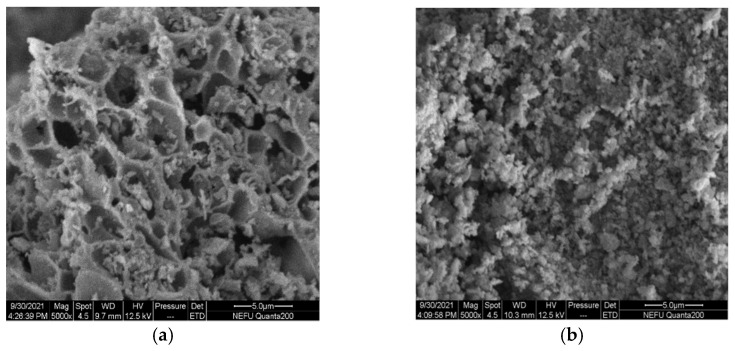
SEM of catalyst samples: (**a**) La_0.8_Ce_0.2_CoO_3_; and (**b**) La_0.8_Ca_0.2_CoO_3_.

**Figure 11 materials-15-04149-f011:**
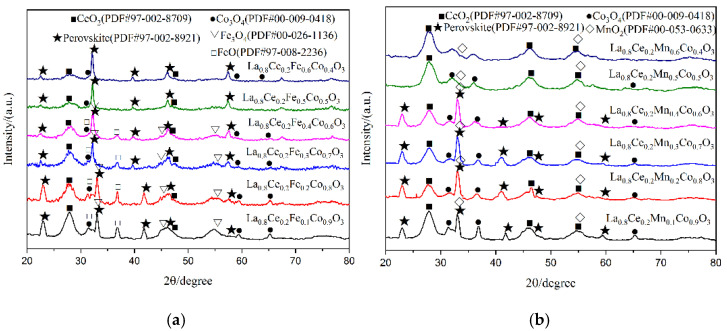
XRD patterns of catalyst samples: (**a**) La_0.8_Ce_0.2_Co_1−y_Fe_y_O_3_ catalyst samples; and (**b**) La_0.8_Ce_0.2_Co_1−y_Mn_y_O_3_ catalyst samples.

**Figure 12 materials-15-04149-f012:**
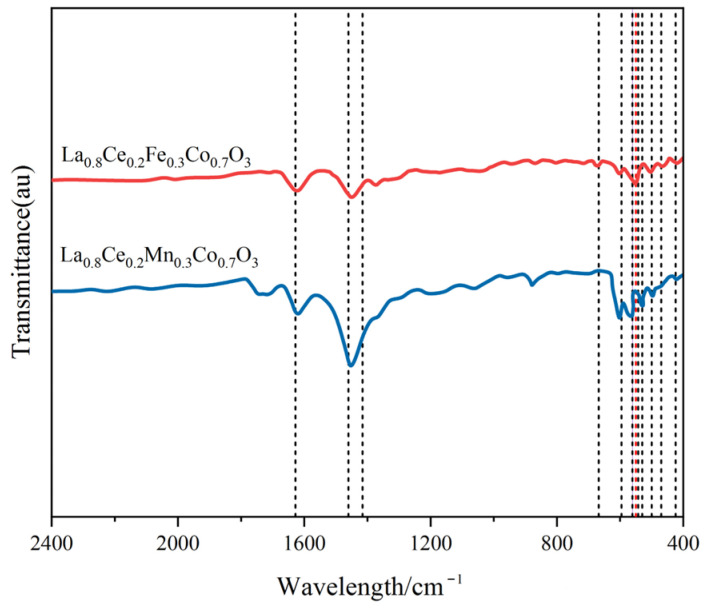
FTIR spectra of La_0.8_Ce_0.2_Fe_0.3_Co_0.7_O_3_ and La_0.8_Ce_0.2_Mn_0.3_Co_0.7_O_3_ catalyst samples.

**Figure 13 materials-15-04149-f013:**
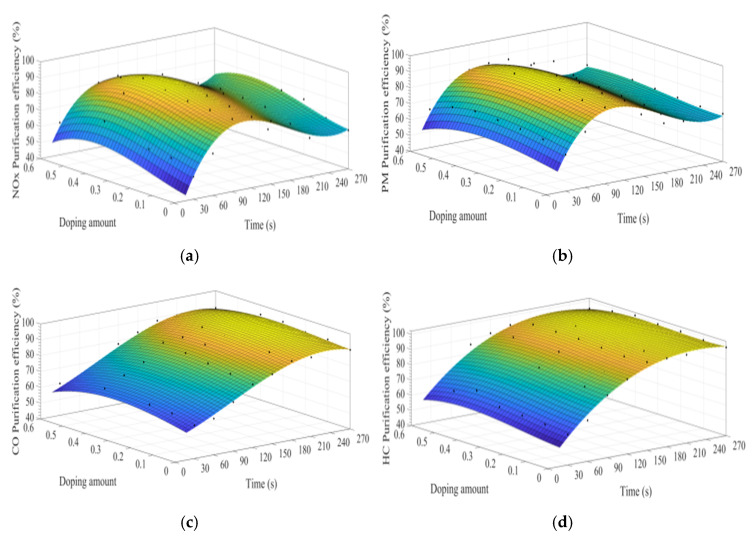
Purification efficiency curve of catalyst when B site was doped with iron ions: (**a**) NO_x_; (**b**) PM; (**c**) CO; and (**d**) HC.

**Figure 14 materials-15-04149-f014:**
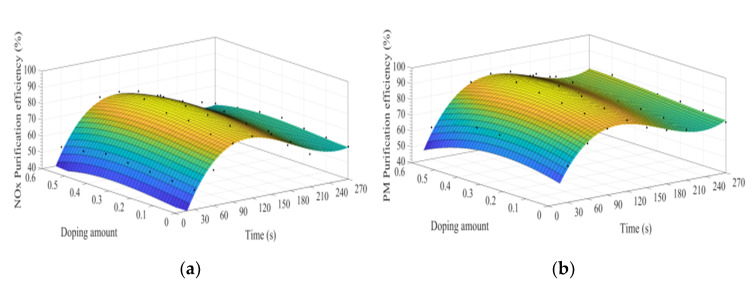
Purification efficiency curve of catalyst when B site was doped with manganese ions: (**a**) NO_x_; (**b**) PM; (**c**) CO; and (**d**) HC.

**Figure 15 materials-15-04149-f015:**
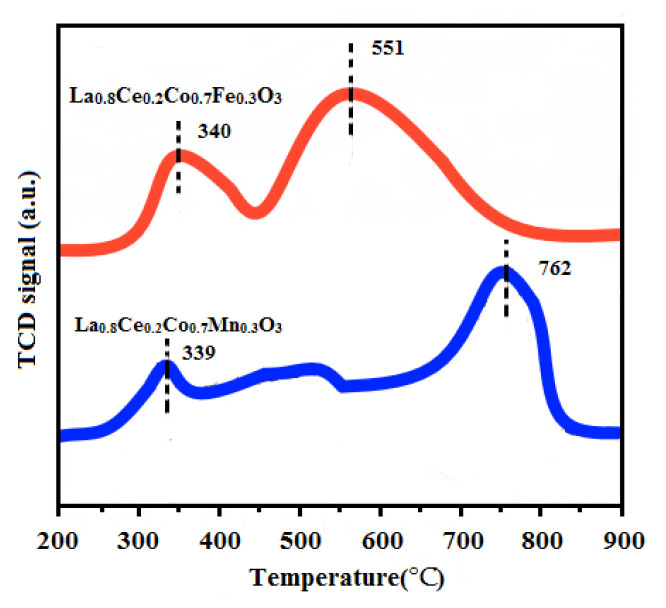
H_2_-TPR profiles of La_0.8_Ce_0.2_Co_0.7_Fe_0.3_O_3_ and La_0.8_Ce_0.2_Co_0.7_Mn_0.3_O_3._

**Figure 16 materials-15-04149-f016:**
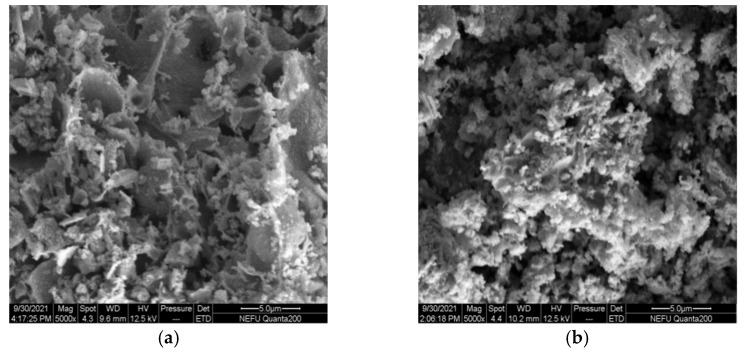
SEM of catalyst samples: (**a**) La_0.8_Ce_0.2_Co_0.7_Fe_0.3_O_3_; and (**b**) La_0.8_Ce_0.2_Co_0.7_Mn_0.3_O_3_.

**Figure 17 materials-15-04149-f017:**
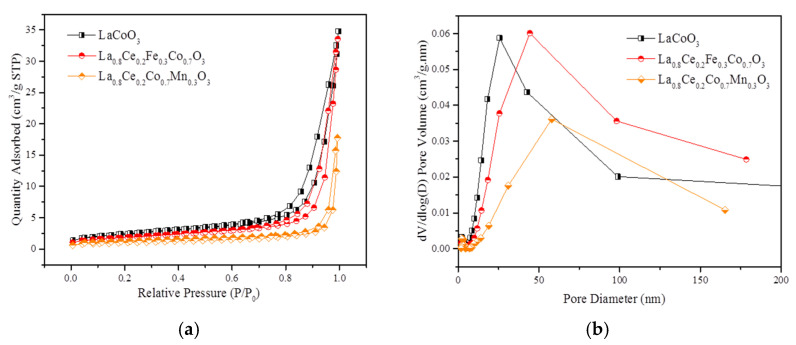
N_2_ adsorption isotherms and PSD curves of catalyst samples: (**a**) N_2_ adsorption isotherms; and (**b**) PSD curves.

**Table 1 materials-15-04149-t001:** Concentration of simulated exhaust gas pollutants.

Gas Composition	Concentration
O_2_	10%
C_2_H_4_	300 ppm
CO	800 ppm
NO	800 ppm
N_2_	dilution gas

**Table 2 materials-15-04149-t002:** The Textural parameters of catalyst samples.

Samples	BET Surface Area (m^2^/g)	Average Pore Diameter (nm)
LaCoO_3_	8.19	26.75
La_0.8_Ce_0.2_Fe_0.3_Co_0.7_O_3_	6.38	34.50
La_0.8_Ce_0.2_Co_0.7_Mn_0.3_O_3_	3.63	39.64

**Table 3 materials-15-04149-t003:** The purification data for the best samples (x = 0.2; y = 0; 0.3).

	Doping Amount	NO_x_	PM	HC	CO
Purification Data(%)					
M = Ce; x = 0.2; y = 0		85	90	94	100
M = Ca; x = 0.2; y = 0		72	90	94	100
M = Ce; N = Mn; x = 0.2; and y = 0.3		87	96	94	100
M = Ce; N = Fe; x = 0.2; and y = 0.3		92	95	94	100

## Data Availability

Data can be made available upon request.

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
