# Peer review of "Study of Ce, Ca, Fe, and Mn-Doped LaCoO3 Perovskite Oxide for the Four-Way Purification of PM, NOx, CO, and HC from Diesel Engine Exhaust"

_materials, 2022, doi:10.3390/ma15124149_

Round 1

Reviewer 1 Report

Dear Authors! Thank you for your manuscript, submitted to Materials. I have read it with pleasure. I think, it includes the meaningful and useful information about the purification of diesel engine exhaust by perovskite oxides, based on doped-LaCoO3. Meanwhile, I have some comments, that would be corrected.

  1. Lines 17, 247. "..samples produce small distortion".  What is the mentioned distortion? Crystal structure distortion or not?
  2. Line 18. "However, .." What is the contradiction with previous text? It is not clear. Similarly, Line 22.  Please, correct the sentences.
  3. Please, explain the reasons for the choice of dopants for LaCoO3: Ca, Ce, Fe, Mn. Introduction Part did not answer this question.
  4. If the samples were obtained with sol-gel method, why was not ethyleneglycol used? Was it real reaction of polycondensation (Line 119) without ethyleneglycol using citric acid only?
  5. How the doped-LaCoO3 powder samples were applied on cordierite support (what was the experimental method used)? 
  6. Where did the data for ionic radii obtain from? Please, add the reference. Or, it is wishable, please, use data of Shannon (Acta Cryst. 1976, A32, 751) reported for La+3, Ca+2, Ce+4 (XII) cations.
  7. Fig 4a, XRD patterns for the samples with Ce-content >0.3. It is seen clearly, that these samples did not form a perovskite phase, because the XRD patterns did not have peaks near 33o.
  8. Fig 4b. Some Ca-samples included CaCO3. Was it calcium carbonate? Probably, it was calcium cobaltite CaCoO3, was not it?
  9. Lines 215, 298. Please, justify the presence of hydroxyl and carbonate groups.
  10. It is useful to add the Table with purification data for the best samples (x=0.2; y=0; 0.3) and complete it with literature data for comparison.
  11. Is it possible to make a correlation between pyrification degree and crystallite sizes? 
  12. Line 281. The conclusion about various doping levels of LaCoO3 with Mn and Fe needs in clearification, because, due to literature data,  there are continuous series of solid solutions in LaCoO3-LaMnO3 and LaCoO3-LaFeO3 binary systems both. 
  13. Line 283. What is unusual in structures of mentioned samples (x=0.2; y=0.3), that could be responsible for the best catalyst properties, considering their multi-phase character?
  14. What were the experimental errors for purification data?

Author Response

Thank you for your letter and for the reviewer’s comments concerning our manuscript entitled “Four-way Purification of Diesel Engine Exhaust by Doping A and B Sites in LaCoO3 Perovskite Oxide”. We have revised our manuscript accordding to reviewer’s comments, and the revised portion are marked in the paper. The revised article has become better. The main responds to the reviewer’s comments are as follows:

Point 1: Lines 17, 247. "..samples produce small distortion". What is the mentioned distortion? Crystal structure distortion or not?

Response 1: Thanks for reviewer’s comment. According to the characterization of  La1-xMxCoO3(M=Ce; x=0,0.1,0.2,0.3,0.4,0.5,0.6)  by XRD and FT-IR, a small amount cerium ions can be dissolved in the LaCoO3 lattice. However, The effective ionic radius of lanthanum ions and cerium ions are 1.10 Å and 0.87 Å respectively, this leads to lattice distortion. It can be observed from Figure 5(a) and (b), the characteristic diffraction peaks of LaCoO3 doping with cerium ions and calcium ions near2θ=23.3° moved left slightly. And it can be observed from Figure 10(a) and (b), the characteristic diffraction peaks of La0.8Ce0.2CoO3 doping with iron ions and manganese ions near 2θ=33°moved right slightly. We have added an analysis of shift in the main diffraction peak position, and confirmed the accommodation of these compounds within the perovskite lattice.

Point 2: Line 18. "However, .." What is the contradiction with previous text? It is not clear. Similarly, Line 22.  Please, correct the sentences.

Response 2: Thanks for reviewer’s comment. We have corrected the sentences.

Point 3: Please, correct the sentences. Please, explain the reasons for the choice of dopants for LaCoO3: Ca, Ce, Fe, Mn. Introduction Part did not answer this question.

Response 3: Thanks for reviewer’s comment. We have explained and added the reasons for the choice of dopants for LaCoO3: Ca, Ce, Fe, Mn in the section of “Introduction”.

Point 4: If the samples were obtained with sol-gel method, why was not ethyleneglycol used? Was it real reaction of polycondensation (Line 119) without ethyleneglycol using citric acid only?

Response 4: Thanks for reviewer’s comment. It was real reaction of polycondensation without ethyleneglycol using citric acid only. Before samples preparation, we consulted the relevant literature and obtained that citric acid can be regarded as polycarboxylic acid, which can form 1:1 M-Y complex with most metal ions. Citric acid has hydroxyl and carboxyl groups, which can be polymerized into polymers by esterification reaction[1,2]. Therefore, we used excess citric acid only when preparing the sample. In future research, we will carry out further research according to your comments.

[1] Meng, F., Song, Y., Li, X., Cheng, Y., & Li, Z. Catalytic methanation performance in a low-temperature slurry-bed reactor over Ni–ZrO2 catalyst: effect of the preparation method. Journal of Sol-Gel Science and Technology, 2016, 80(3), 759–768.

[2] Natile, M. M.; Ugel, E.; Maccato, C.; Glisenti, A. LaCoO3: Effect of synthesis conditions on properties and reactivity. Applied Catalysis B: Environmental 2007, 72(3-4), 351–362.

Point 5: How the doped-LaCoO3 powder samples were applied on cordierite support (what was the experimental method used)?

Response 5: Thanks for reviewer’s comment. The moderate PM and catalyst samples were set in the circulating blower, which was sealed before the supporting process. When the circulating blower was started, the porous structure of cordierite can capture all of the powder PM and catalyst samples. we have added the experimental method in the section of “Experimental system and methods”.

Point 6: Where did the data for ionic radii obtain from? Please, add the reference. Or, it is wishable, please, use data of Shannon (Acta Cryst. 1976, A32, 751) reported for La+3, Ca+2, Ce+4 (XII) cations.

Response 6: Thanks for reviewer’s comment. We have added the reference, and revised the data for ionic radii in the section of “Results of characterization on catalyst samples doping at A site”.

Point 7: Fig 4a, XRD patterns for the samples with Ce-content >0.3. It is seen clearly, that these samples did not form a perovskite phase, because the XRD patterns did not have peaks near 33o.

Response 7: Thanks for reviewer’s comment. When Ce-content >0.3, excessive cerium ions and cobalt ions cannot be completely dissolved in the LaCoO3 lattice, this leads to a large number of impurities in the prepared samples, for example CeO2 and Co3O4.Therefore, the XRD characteristic diffraction peaks of LaCoO3 near 2θ= 33.0° was extraordinary inconspicuous. We have added the above explanation in the section of “Results of characterization on catalyst samples doping at A site”.

Point 8: Fig 4b. Some Ca-samples included CaCO3. Was it calcium carbonate? Probably, it was calcium cobaltite CaCoO3, was not it?

Response 8: Thanks for reviewer’s comment. When we analysed the impurity phase in the La0.8Ca0.2CoO3 catalyst samples, the FT-IR spectra of La0.8Ca0.2CoO3 catalyst samples was combined. We could only find the bending vibration of Co–O bonding in the BO6 octahedron, and the vibration band of Co–O, which were attributed to Co3O4. The FT-IR spectra of La0.8Ca0.2CoO3 catalyst samples  appeared the absorption peaks of 875 and 712 cm−1 belong to the calcite crystals, which were related to the bending vibration of C–O bond. Therefore, we guessed the small amount of impurity phase included CaCO3. Moreover, under a vacuum-sintered process, the carbonate is harder to eliminate[3].

[3] Sekak, K. A., & Lowe, A, Structural and Thermal Characterization of Calcium Cobaltite Electrospun Nanostructured Fibers. Journal of the American Ceramic Society, 2010, 94(2), 611–619.

Point 9: Lines 215, 298. Please, justify the presence of hydroxyl and carbonate groups.

Response 9: Thanks for reviewer’s comment. We have justified the presence of carbonate groups. (see question 8). And we have added the explanation that the presence of hydroxyl in the section of “Results of characterization on catalyst samples doping at A site”.

Point 10: It is useful to add the Table with purification data for the best samples (x=0.2; y=0; 0.3) and complete it with literature data for comparison.

Response 10: Thanks for reviewer’s comment. We have added the Table with purification data for the best samples (x=0.2; y=0; 0.3) and complete it with literature data for comparison.

Point 11: Is it possible to make a correlation between purification degree and crystallite sizes?

Response 11: Thanks for reviewer’s comment. We have added the correlation between pyrification degree and crystallite sizes in the section of “Results of Simulation experiments on catalyst samples doping at B site”.

Point 12: Line 281. The conclusion about various doping levels of LaCoO3 with Mn and Fe needs in clearification, because, due to literature data,  there are continuous series of solid solutions in LaCoO3-LaMnO3 and LaCoO3-LaFeO3 binary systems both.

Response 12: Thanks for reviewer’s comment. We have clarified the conclusion about various doping levels of  LaCoO3 with Mn and Fe in the section of “Results of characterization on catalyst samples doping at B site”.

Point 13: Line 283. What is unusual in structures of mentioned samples (x=0.2; y=0.3), that could be responsible for the best catalyst properties, considering their multi-phase character?

Response 13: Thanks for reviewer’s comment. The catalytic activity of perovskite-type catalysts for purifying the pollutants emitted by the engine can be enhanced by substituting the A-site and B-site reasonably[4]. The cerium ions doping of the A site helps to increase the activity of the catalyst lattice oxygen, and make the B site ions exist in multiple valence states, causing the increases of oxygen vacancy in the catalyst. The valence states of cerium ions and iron ions in La0.8Ce0.2Fe0.3Co0.7O3 can achieve the transition between +3 and +4. Therefore, La0.8Ce0.2Fe0.3Co0.7O3 samples has excellent oxygen storage and release capacity, which can inhibit the oxidation process of PM, HC, CO and reduction process of NOx.

[4] L. Guo, L. Bo, Y. Li, et al. Sr doping effect on the structure property and NO oxidation performance of dual-site doped perovskite La(Sr)Co(Fe)O3, Solid State Sciences, 2021, vol. 113.

Point 14: What were the experimental errors for purification data?

Response 14: Thanks for reviewer’s comment. Experimental errors for purification data include systematic error and accidental error. Systematic error caused by inherent accuracy of flowmeter, gas analyzer, electronic balance, water bath pot and other test instrument. Accidental error caused by the accidental fluctuation of air pressure, temperature and humidity in the test environment to the test instrument.

Reviewer 2 Report

This manuscript investigates the four way purification of PM, NOx, CO and HC emitted by diesel exhaust over several La1-xMxCo1-yNyO3 (with M= Ce or Ca and N=Mn or Fe) perovskite-type catalysts. These catalysts were characterized by XRD, FTIR and SEM, whereas their catalytic behaviour was evaluated by response surfaces. It is proved by the authors that A and B cations doping promotes the four-way purification efficiency. This promoting effect is more noticeable when La3+ and Co3+ are substituted by Ce and Fe ions, respectively. This behaviour was attributed to doping agent better accommodation within the perovskite lattice as well as oxidation state. However, no enough evidences about these aspects are included throughout the manuscript. Thus, several aspects require careful consideration and revision before the manuscript being suitable for its publication:

1.       The title is not enough specific, maybe the doping elements should be defined. Thus, the title must be reconsidered.

2.       The introduction does not follow coherent structure and lead to confusion about the real motivation of the manuscript.

3.       The general nomenclature should be as follows: La1-xMxCo1-yNyO3 instead of La1-xMxNyCo1-yO3, since N element is substituting Co.

4.       There are several grammar, spelling and writing mistakes that should be corrected throughout the manuscript. Some examples bellow:

Line 33:  seriously damages the respiratory system of human body, and even leads to the increase.

Line 43:  citric acid-ligated impregnation method (is no proper definition).

Line 45: Cs+

Lines 308 and 317: subscripts in La0.8Ce0.2CoO3

Line 326: sintered

Line 327:….samples exist porous structure, The pore

Experimental
==========

1.       In the introduction, it is suggested that one of the main purposes of this study is the simultaneous purification of CO, HC, NOx and PM emitted by diesel engines. However, this study is only carried out at 600 °C, which limits the evaluation of the real applicability of these materials. In fact, typical of modern, fuel efficient, light duty diesel engines tend to operate below 350 °C (for example, typical FTP-75 temperatures range from ca. 180 - 350 C; European driving cycles tend to be even cooler). To my mind, the results in this manuscript indicate that perovskites might be used useful for improving high temperature activity (>350 °C) but that the use of a precious metal will be required to increase the low temperature activity (Industrial & Engineering. Chemistry Research 2021, 60, 18, 6525–6537).

2.       How can you operate the diesel system for maintaining this narrow set of conditions for a vehicle that operates under a variety of speed and loads? How can you ensure 600 °C driving conditions? The study should be carried under widen operational conditions.

3.       The numeration in the Figure 2 footnote is wrong.

4.       The explanation of the reaction system is not clear. Further information should be included about the reactor system, operational conditions (space time and space velocity) and gas composition analysis (method and equipment).

5.       Is it used a carried gas during activity test?

Methodology
===========

1.    What the authors refer to simulation experiments? How they obtain Figure 6, 7, 11 and 12? This information is missing in the experimental section.

2.    These figures are response surfaces obtained from different experimental results?

Results
=============

1.     The authors show some differences in doping agent’s accommodation by XRD analysis. Is it observed any shift in the main diffraction peak position as long as La or Co are partially substituted by Ce, Ca, Mn or Fe? This information would confirm the accommodation of these compounds within the perovskite lattice

2.     Why not information about textural properties in included throughout the Manuscript? It is widely accepted that the impurities formation as a consequence of the limited accommodation of A and B cations leads to a progressive decrease of specific surface area (Molecular Catalysis, 488, 2020, 110913)

3.     The authors suggest that B cation oxidation state or oxygen vacancies formation are the key factors to explain during reduction/oxidation reactions. However, no evidences are included to support this aspect. Temperature programmed techniques, such as O2-TPD or H2-TPR experiments, will provide specific information about the redox properties and oxygen mobility of the different samples.

4.     The interpretation of Figures 6, 7, 11 and 12 will be easier, if isocurves corresponding to different levels of NOx conversion projected to the A/B cation substitution degree-time spaces are included.

5.     The authors suggest that Co3+ change to Co4+ by La3+ substitution. However, the formation of oxygen vacancies, instead of cobalt oxidation state modification, it is more widely reported in the literature (Chemical Engineering Journal, 428, 2022, 131352). How the authors support their assumption?

6.     How stable and sustainable are the catalyst materials? Some cyclic short term aging studies should be performed since deactivation of any one component will disrupt the entire performance.

7.     The main point is that water is present in the feed, and it is well-known that it has an influence on the storage/reduction capacities of the catalysts; have the authors an opinion on this? Do they consider expanding and completing their studies considering also this aspect??

Author Response

Thank you for your letter and for the reviewer’s comments concerning our manuscript entitled “Four-way Purification of Diesel Engine Exhaust by Doping A and B Sites in LaCoO3 Perovskite Oxide”. We have revised our manuscript accordding to reviewer’s comments, and the revised portion are marked in the paper. The revised article has become better. The main responds to the reviewer’s comments are as follows:

Point 1: The title is not enough specific, maybe the doping elements should be defined. Thus, the title must be reconsidered.

Response 1: Thanks for reviewer’s comment. We have revised the title to “Study of Ce, Ca, Fe, Mn-doped LaCoO3 Perovskite Oxide for the Four-way Purification of PM, NOx, CO and HC from Diesel Engine Exhaust according to your comments.

Point 2: The introduction does not follow coherent structure and lead to confusion about the real motivation of the manuscript.

Response 2: Thanks for reviewer’s comment. We have added a summary of the literature review in the second paragraph of the introduction. On this basis, we further clarify the research motivation of our manuscript.

Point 3: The general nomenclature should be as follows: La1-xMxCo1-yNyO3 instead of La1-xMxNyCo1-yO3, since N element is substituting Co.

Response 3: Thanks for reviewer’s comment. We have replaced all La1-xMxNyCo1-yO3 in the manuscript with La1-xMxCo1-yNyO3.

Point 4: There are several grammar, spelling and writing mistakes that should be corrected throughout the manuscript. Some examples bellow:

Line 33:  seriously damages the respiratory system of human body, and even leads to the increase.

Line 43:  citric acid-ligated impregnation method (is no proper definition).

Line 45: Cs+

Lines 308 and 317: subscripts in La0.8Ce0.2CoO3

Line 326: sintered

Line 327:….samples exist porous structure, The pore

Response 4: Thanks for reviewer’s comment. We have corrected the grammar, spelling and writing mistakes.

Point 5: In the introduction, it is suggested that one of the main purposes of this study is the simultaneous purification of CO, HC, NOx and PM emitted by diesel engines. However, this study is only carried out at 600 °C, which limits the evaluation of the real applicability of these materials. In fact, typical of modern, fuel efficient, light duty diesel engines tend to operate below 350 °C (for example, typical FTP-75 temperatures range from ca. 180 - 350 C; European driving cycles tend to be even cooler). To my mind, the results in this manuscript indicate that perovskites might be used useful for improving high temperature activity (>350 °C) but that the use of a precious metal will be required to increase the low temperature activity (Industrial & Engineering. Chemistry Research 2021, 60, 18, 6525–6537).

Response 5: Thanks for reviewer’s comment. In this study, we did not consider the influence of different diesel engine exhaust temperatures on perovskite materials. In the next engine bench test, we will carry out the research according to the typical FTP-75 and the use of a precious metal in perovskites. The suggestions mentioned from comments will be of great help to our future research, which will promote the wide application of perovskite materials under different engine conditions.

Point 6: How can you operate the diesel system for maintaining this narrow set of conditions for a vehicle that operates under a variety of speed and loads? How can you ensure 600 °C driving conditions? The study should be carried under widen operational conditions.

Response 6: Thanks for reviewer’s comment. According to the research “NO2-Assisted Soot Regeneration Behavior in a Diesel Particulate Filter with Heavy Duty Diesel Exhaust Gases”, the exhaust gas temperature of diesel engine is is allowed to vary from 550-650 °C[1]. Meanwhile, in order to simplify the test process, we chose the operating condition of 600 °C. In the future research, we will take the operating temperature conditions as the main variable to study the purification of automobile exhaust by perovskite materials.

[1] Kim, J. H.; Kim, M. Y.; Kim, H. G. NO2-Assisted Soot Regeneration Behavior in a Diesel Particulate Filter with Heavy-Duty Diesel Exhaust Gases. Numerical Heat Transfer, Part A: Applications2010, 58(9), 725–739.

Point 7: The numeration in the Figure 2 footnote is wrong.

Response 7: Thanks for reviewer’s comment. We have revised the numeration in the Figure 2 footnote.

Point 8: The explanation of the reaction system is not clear. Further information should be included about the reactor system, operational conditions (space time and space velocity) and gas composition analysis (method and equipment).

Response 8: Thanks for reviewer’s comment. We have added gas composition and space velocity in the section of “The explanation of the reaction system” and further explained the flow chart of the test system.

Point 9: Is it used a carried gas during activity test?

Response 9: Thanks for reviewer’s comment. A carried gas was used during the process of activity test.

Point 10: What the authors refer to simulation experiments? How they obtain Figure 6, 7, 11 and 12? This information is missing in the experimental section.

Response 10: Thanks for reviewer’s comment. In order to save the experimental cost and facilitate the control of influence parameters. During the simulation experiments, the simulated exhaust gas can be provided by the gas cylinder group (O2, C2H4, N2, CO and NO) and mixed by the mixing chamber [2,3]. The tube furnace was used to control the temperature of the mixed gas and the water-bath pot can maintain the exhaust temperature constant. The electronic balance and the gas analyzer were used to measure the mass of PM and analyze the exhaust gas composition respectively. Simulation test is a scientific test conducted by simulating engine bench test. We simulated some engine bench tests through simulation experiments.

In order to explain the simulation test process more clearly, we added the simulation test flow chart in the experimental section.

[2] Wooseok Kang, Byungchul Choi, Hwanam Kim. Characteristics of the simultaneous removal of PM and NOx using CuNb-ZSM-5 coated on diesel particulate filter. Journal of Industrial and Engineering Chemistry. 2013,19(4):1406-1412.

[3] Danfeng D, Yinghui W, Xiurong G. Experiment on the performance of the NTP synergisticwood fiber diesel exhaust purifier. J Harbin Eng Univ. 2019, 40(02):419–425.

Point 11: These figures are response surfaces obtained from different experimental results?

Response 11: Thanks for reviewer’s comment. In the process of processing experimental data, time and doping amount were taken as independent variables, and the purification efficiency of NOx, PM, CO and HC were taken as dependent variables. In order to analyze the regression relationship between experimental variables and experimental indicators, Pictures of response surface are obtained from different test data.

Point 12: The authors show some differences in doping agent’s accommodation by XRD analysis. Is it observed any shift in the main diffraction peak position as long as La or Co are partially substituted by Ce, Ca, Mn or Fe? This information would confirm the accommodation of these compounds within the perovskite lattice.

Response 12: Thanks for reviewer’s comment. It can be observed from Figure 5(a) and (b), the characteristic diffraction peaks of LaCoO3 doping with cerium ions and calcium ions near2θ=23.3° moved left slightly. And it can be observed from Figure 10(a) and (b), the characteristic diffraction peaks of La0.8Ce0.2CoO3 doping with iron ions and manganese ions near 2θ=33° moved right slightly.We have added an analysis of shift in the main diffraction peak position, and confirmed the accommodation of these compounds within the perovskite lattice.

Point 13: Why not information about textural properties in included throughout the Manuscript? It is widely accepted that the impurities formation as a consequence of the limited accommodation of A and B cations leads to a progressive decrease of specific surface area (Molecular Catalysis, 488, 2020, 110913)

Response 13: Thanks for reviewer’s comment. We have added the information about textural properties in the section of “Results of Simulation experiments on catalyst samples doping at B site”. And the N2 adsorption isotherms and PSD curves of catalyst samples were further analyzed.

Point 14: The authors suggest that B cation oxidation state or oxygen vacancies formation are the key factors to explain during reduction/oxidation reactions. However, no evidences are included to support this aspect. Temperature programmed techniques, such as O2-TPD or H2-TPR experiments, will provide specific information about the redox properties and oxygen mobility of the different samples.

Response 14: Thanks for reviewer’s comment. It is widely accepted that in the LaCoO3-based perovskite, the reversible oxidation state of Co3+/Co2+ is a key factor for the oxidation reactions and Sr doping generates oxygen vacancies, which could further facilitate the mobility of bulk oxygen[4-6]. Some studies suggested that the adsorption of oxygen species over Co3+ sites and oxygen vacancies in perovskite structures play an important role in the CO oxidation[7]. In the LaMnO3 catalyst, the partial substitution of  K+ into A site La3+ enhanced the catalytic activity. The first one is that A site cation (La3+) were replaced partly by K+ and partial Mn3+ changed to Mn4+, which had better catalytic oxidation activity than Mn3+. The second one is the increase in the content of oxygen vacancy (V0) in La1-xKxMnO3, which increases the adsorption and activation oxygen at catalyst surface. Therefore, it can improve oxidant activity[8]. In order to simplify the test process, we chose the operating condition of 600 °C in this paper. In the future research, we will take the operating temperature conditions as the main variable to study the purification of automobile exhaust by perovskite materials. And the temperature programmed techniques will be used to explore the the redox properties and oxygen mobility of different samples.

[4] X.-G. Li, Y.-H. Dong, H. Xian, W.Y. Hernández, M. Meng, H.-H. Zou, A.-J. Ma, T.-Y. Zhang, Z. Jiang, N. Tsubaki, P. Vernoux, De-NOx in alternative lean/rich atmospheres on La1−xSrxCoO3  perovskites, Energ. Environ. Sci. 4 (2011) 3351.

[5] S. Ponce, M.A. Pe˜na, J.L.G. Fierro, Surface properties and catalytic performance in methane combustion of Sr-substituted lanthanum manganites, Appl. Catal. B-Environ. 24 (2000) 193–205.

[6] C. Zhang, W. Hua, C. Wang, Y. Guo, Y. Guo, G. Lu, A. Baylet, A. Giroir-Fendler, The effect of A-site substitution by Sr, Mg and Ce on the catalytic performance of LaMnO3 catalysts for the oxidation of vinyl chloride emission, Appl. Catal. B-Environ. 134-135 (2013) 310–315.

[7] Omata K, Takada T, Kasahara S, Yamada M. Active site of substituted cobalt spinel oxide for selective oxidation of CO/H-2.2. Appl Catal A.1996,146:255–267.

[8] Hong W, Zhen Z, Xu C M, et al. Nanometric La1−xKxMnO3, Perovskite-type oxides highly active catalysts for the combustion of diesel soot particle under loose contact conditions. Catalysis Letters, 2005, 102(3):251-256.

Point 15: The interpretation of Figures 6, 7, 11 and 12 will be easier, if isocurves corresponding to different levels of NOx conversion projected to the A/B cation substitution degree-time spaces are included.

Response 15: Thanks for reviewer’s comment. Your suggestion is very enlightening to me. We will certainly use your method in the future experimental data processing.

Point 16: The authors suggest that Co3+ change to Co4+ by La3+ substitution. However, the formation of oxygen vacancies, instead of cobalt oxidation state modification, it is more widely reported in the literature (Chemical Engineering Journal, 428, 2022, 131352). How the authors support their assumption?

Response 16: Thanks for reviewer’s comment. The introduction of Cu would promote the generation of oxygen vacancies in LaCoO3 in the literature (Chemical Engineering Journal, 428, 2022, 131352), and the Co2+,Co3+, Cu+ and Cu2+ coexisted on the surface of LCC-4 [9]. In this paper, the introduction of Ca2+ can change Co3+ to Co4+ due to the principle of electric neutrality, the introduction of Ce4+ can change Co3+ to Co2+ due to the principle of electric neutrality. That would promote the generation of oxygen vacancies in LaCoO3. Therefore, the principle of oxygen vacancies formation in this paper is similar to that in the literature (Chemical Engineering Journal, 428, 2022, 131352).

[9] Xie, L., Liu, X., Chang, J., Zhang, C., Li, Y., Zhang, H., Zhan, S., & Hu, W. Enhanced redox activity and oxygen vacancies of perovskite triggered by copper incorporation for the improvement of electro-Fenton activity. Chemical Engineering Journal, 2022, 428, 131352.

Point 17: How stable and sustainable are the catalyst materials? Some cyclic short term aging studies should be performed since deactivation of any one component will disrupt the entire performance.

Response 17: Thanks for reviewer’s comment. La-based perovskite oxides have showed high thermal stability and good redox activity in many studies[10-12]. In the next engine bench test, We will conduct experimental research on the service life of perovskite catalytic materials, with a view to practical application.

[10] S.A. Hosseini, B. Mehri, A. Niaei, B. Izadkhah, C. Alvarez-Galvan, J.G.L. Fierro, Selective catalytic reduction of NO x  by CO over LaMnO 3  nano perovskites prepared by microwave and ultrasound assisted sol–gel method, J. Sol-Gel Sci. Technol. 2018 , 85, 647–656.

[11] R. Zhang, A. Villanueva, H. Alamdari, S. Kaliaguine, Reduction of NO by CO over nanoscale LaCo1−xCuxO3 and LaMn1−xCuxO3 perovskites, J. Mol. Catal. A Chem. 2006, 258, 22–34.

[12] M.A. Pena, J.L.G. Fierro, Chemical structures and performance of perovskite oxides, Chem. Rev. 2001,101, 1981–2018.

Point 18: The main point is that water is present in the feed, and it is well-known that it has an influence on the storage/reduction capacities of the catalysts; have the authors an opinion on this? Do they consider expanding and completing their studies considering also this aspect?

Response 18: Thanks for reviewer’s comment. Diesel exhaust gas mainly contains PM, NOx, CO, HC and so on. However, humidity has an influence on the storage/reduction capacities of the catalysts[13]. In the design of this simulation test, the tube furnace is used to heat the simulated gas. We considered that there will be explosion and other hazards when the tubular furnace heats water. And humidity is not easy to control in this simulation test bench. If we expand this study, a new simulation test bench needs to be designed and built. However, this study cannot be completed in a short time(10 days). In our future research, we will take the humidity as the main influence factor to complete our studies.

[13] Onrubia-Calvo, J. A., Pereda-Ayo, B., Cabrejas, I., De-La-Torre, U., & González-Velasco, J. R. Ba-doped vs. Sr-doped LaCoO3 perovskites as base catalyst in diesel exhaust purification. Molecular Catalysis, 2020, 488, 110913.

Round 2

Reviewer 1 Report

Dear Authors! Thank you for your attention to my comments and to comments of another Reviewer. The manuscript was improved, no doubt. Nevertheless, I would like to notice, that Introduction Part could be expanded more with relevant references, concerning to last years. And manuscript needs in strong proofreading - the creation of MDPI template resulted in spaces disappearance - the text became partially merged . 

Author Response

Thank you for your letter and for the reviewer’s comments concerning our revised manuscript entitled “Four-way Purification of Diesel Engine Exhaust by Doping A and B Sites in LaCoO3 Perovskite Oxide”. We believe that the comments have been highly constructive and very useful to restructure the manuscript. We have revised our manuscript accordding to reviewer’s comments, and the revised portion are marked in the paper. The revised article has become better. The main responds to the reviewer’s comments are as follows:

Point 1: The manuscript was improved, no doubt. Nevertheless, I would like to notice, that Introduction Part could be expanded more with relevant references, concerning to last years. And manuscript needs in strong proofreading - the creation of MDPI template resulted in spaces disappearance - the text became partially merged . 

Response 1: Thanks for reviewer’s comment. We have expanded the Introduction Part with relevant references. And our manuscript has been proofread accordding to the MDPI template.

Reviewer 2 Report

The manuscript still presents several grammar, spelling and writing mistakes.

Although some improvements have been carried out, the manuscript can still be greatly improved. Some examples below:

1. No description of BET analysis has been included.

2. The total flow rate should be specified.

3. It is difficult to evaluate Figures 7, 8, 12, and 13 in the current form. The heigh of the conversion axis should be increased to improve their resolution.

4. Line 238. The oxidation state of cerium should be 3+ or 4+. Thus, the cobalt oxidation state should be 2+/3+ not 4+.

5. The text format throughout the manuscript is not homogeneous.

6. Based on the reaction pathway reported in Equations 5-19, the redox properties and the oxygen mobility of the catalysts should be a key factor to explain the differences in conversion efficiencies Thus, XPS, H2-TPR and/or O2-TPD experiments should be carried out.

Author Response

Thank you for your letter and for the reviewer’s comments concerning our revised manuscript entitled “Study of Ce, Ca, Fe, Mn-doped LaCoO3 Perovskite Oxide for the Four-way Purification of PM, NOx, CO and HC from Diesel Engine Exhaust”. We believe that the comments have been highly constructive and very useful to restructure the manuscript. We have revised our manuscript accordding to reviewer’s comments, and the revised portion are marked in the paper. The revised article has become better. The main responds to the reviewer’s comments are as follows:

Point 1: The manuscript still presents several grammar, spelling and writing mistakes.

Response 1: Thanks for reviewer’s comment. We have revised several grammar, spelling and writing mistakes.

Point 2: No description of BET analysis has been included.

Response 2: Thanks for reviewer’s comment. We have added the description of BET analysis in the section of “Characterization” and “Results of Simulation experiments on catalyst samples doping at B site”,  which are consistent with the research results of “The impurities formation as a consequence of the limited accommodation of A and B cations leads to a progressive decrease of specific surface area”[1] .

[1] Onrubia-Calvo, J. A.; Pereda-Ayo, B.; Cabrejas, I.; De-La-Torre, U.; González-Velasco, J. R. Ba-doped vs. Sr-doped LaCoO3 perovskites as base catalyst in diesel exhaust purification. Molecular Catalysis 2020, 488, 110913.

Point 3: The total flow rate should be specified.

Response 3: Thanks for reviewer’s comment. We have specified the total flow rate in the section of “Experimental system and methods”.

Point 4: It is difficult to evaluate Figures 7, 8, 12, and 13 in the current form. The heigh of the conversion axis should be increased to improve their resolution.

Response 4: Thanks for reviewer’s comment. We have improved the resolution of Figures 7, 8, 12, and 13 by increasing the heigh of the conversion axis.

Point 5: Line 238. The oxidation state of cerium should be 3+ or 4+. Thus, the cobalt oxidation state should be 2+/3+ not 4+.

Response 5: Thanks for reviewer’s comment. We have revised the oxidation state of cerium and cobalt.

Point 6: The text format throughout the manuscript is not homogeneous.

Response 6: Thanks for reviewer’s comment. We have revised the text format throughout the manuscript accordding to the MDPI template.

Point 7: Based on the reaction pathway reported in Equations 5-19, the redox properties and the oxygen mobility of the catalysts should be a key factor to explain the differences in conversion efficiencies Thus, XPS, H2-TPR and/or O2-TPD experiments should be carried out.

Response 7: Thanks for reviewer’s comment. We have carried out the XPS and H2-TPR analysis in the section of “Results of Simulation experiments on catalyst samples doping at A site” and “Results of Simulation experiments on catalyst samples doping at B site”. Through the analysis above, we explained the differences in conversion efficiencies.
